# A reproducibility study of "Augmenting Genetic Algorithms with Deep Neural Networks for Exploring the Chemical Space"

## Reproducibility Summary

**Scope of Reproducibility**

Nigam et al. (2020) report a genetic algorithm (GA) utilizing the SELFIES representation (Krenn et al., 2020) and also propose an adaptive, neural network-based penalty that is supposed to improve the diversity of the generated molecules. The main claims of the paper are that this GA outperforms other generative techniques (as measured by the penalized logP) and that a neural network-based adaptive penalty increases the diversity of the generated molecules.

**Methodology**

We re-used the code published by the authors after minor refactoring and re-ran the key experiments on a typical workstation (two 16 core Intel Xeon Gold 5218s, Quadro RTX 6000) within two weeks using more recent versions of the dependencies. In particular, we used a new, major version of the SELFIES library and also quantified the diversity of the generated molecules and the effect of different hyperparameters. All of our experiments were tracked on the Weights and Biases platform (Biewald, 2020).[1]

**Results**

Overall, we were able to reproduce comparable results using the SELFIES-based GA—but mostly by exploiting deficiencies of the (easily optimizable) penalized logP fitness function (i.e., generating long, sulfur-containing chains). In addition, we also reproduce results showing that the discriminator can be used to bias the generation of molecules to ones that are similar to the reference set.
Moreover, we propose a new similarity-based adaptive penalty that outperforms the original algorithm on the penalized logP score. We perform ablation studies and show that the complexity of a multilayer network discriminator, as used in the original work, is not crucial to reproduce the dependence of the penalized logP on the penalty weight. Importantly, we emphasize the need for a more representative comparison between algorithms. We analyze the performance of the original algorithm on the Guacamol benchmark set and find that it tends to show low intra-generation diversity.

**What was easy**

Reproducing all the key results (including the plots) was easy since the authors provided code with pre-defined settings and useful comments for every relevant experiment. Hence, it did not require complete implementation from scratch.

**What was difficult**

Without the provided code, reproducing some parts of the papers would have been significantly more time-consuming as the paper did not provide the complete settings required to reproduce the data. In the original article, there was also no indication of how the hyperparameters (e.g., architecture of the discriminator model, weighting of different parts of the fitness function, choice of discriminator loss) were optimized.

---

[1]interactive visualizations are available at `https://bit.ly/3oqhzZl`

## Communication with original authors

We contacted the authors to clarify some questions about discrepancies with the baseline experiment and also provided them with a draft of this report. The original reacted appreciative to the draft of our report.

# 1   Introduction

The accelerated discovery of new materials and molecules requires efficient techniques to explore chemical space. Since chemical design space is vast, simple enumeration and brute-force screening approaches with experimental testing are unfeasible. For example, Polishchuk et al. (2013) estimated the number of drug-like molecules to be between $10^{23}$ and $10^{60}$.

To address this problem, generative techniques such as generative adversarial neural networks (GANs) (Goodfellow et al., 2014; De Cao and Kipf, 2018), variational autoencoders (VAEs) (Kingma and Welling, 2013; Gómez-Bombarelli et al., 2018; Kusner et al., 2017; Liu et al., 2018) and genetic algorithms (GAs) (Devillers, 1996; Supady et al., 2015) have received mounting attention from the chemistry community as means to efficiently perform "inverse design" (Sanchez-Lengeling and Aspuru-Guzik, 2018). That is, rather than exhaustively searching through chemical space, inverse design allows one to optimize chemical design based on a desired target property. A limitation of these approaches, however, lies in the fact that they often use a conventional string-based molecular representations known as simplified molecular-input line-entry notation system (SMILES), which require careful treatment to ensure the validity of the generated molecules, as most SMILES do not correspond to valid molecules. To remedy this problem, Krenn et al. (2020) proposed SELFIES, a string representation for which the random arrangement of characters corresponds to a valid molecules. This property has been recently exploited by Nigam et al. (2021) to efficiently interpolate in chemical space and was also proposed by Nigam et al. (2020) for use in GAs.

Similar to the previous works of Jensen (2019) and Brown et al. (2019), the results by Nigam et al. (2020) suggest that conventional approaches such as GA can outperform generative machine learning models and are therefore an important reference for future advancement in the field.

# 2   Scope of reproducibility

We chose to address the following main claims from the original paper:

**Claim 1: Genetic algorithms operating on SELFIES outperform state-of-the-art generative models**   Nigam et al. showed that their algorithm could achieve higher penalized logP scores than all references they considered. The penalized logP score is hereby defined as logP coefficient penalized with the synthetic accessibility score (SA), and a ring penalty

$$J(m) = \text{logP}(m) - \text{SA}(m) - \text{ring penalty}(m). \tag{1}$$

The logP coefficient is an estimator of the solubility, synthetic accessibility score is a heuristic reported by Ertl and Schuffenhauer (2009) that estimates the ease of synthesis (a lower score means easier synthetic accessibility), and the ring penalty penalizes cycles larger than six. Hence, molecules with higher $J(m)$ are thought to have a more drugable profile than those with low $J(m)$. In addition, Nigam et al. also demonstrated that a GA on SELFIES can perform well on optimization of other objectives such as similarity-constrained optimization or optimization of two conflicting goals.

**Claim 2: A neural network derived discriminator term, $\mathbf{D(m)}$, added to fitness $\mathbf{(J(m) + \beta D(m))}$ increases the diversity of the generated molecules**   Nigam et al. did not quantify the diversity of the generated molecules beyond an unsupervised analysis of the GA trajectory, but achieved higher penalized logP scores than compared to those without the discriminator term—especially when employed in a time-adaptive setting.

In addition to analyzing the reproducibility of these claims, we also attempted to quantify the evolution of the diversity, understand the influence of some hyperparameters, and propose improvements to the adaptive penalty.

# 3   Methodology

We based our replication study on the code provided by Nigam et al.[2] We used one workstation to run all experiments, which are mostly CPU-bound (requiring around $3\,\text{s}$ to $8\,\text{s}$ per GA generation on our workstation). The code we used, including the analysis code, is available in a GitHub repository.[3]

## 3.1   Algorithm

We re-used the GA implementation from the original paper and the discriminator model architecture (two-layer feed-forward network with sigmoid activations trained on the binary cross entropy loss using the Adam optimizer (Kingma

---

[2]http://github.com/aspuru-guzik-group/GA

[3]https://anonymous.4open.science/r/041b3ae6-3bc2-4228-b56f-59e799f2c430/

and Ba, 2015) with learning rate set to $0.001$ and weight decay set to $1 \times 10^{-4}$; model artifacts are available on the Weights and Biases platform). Nigam et al. (2020) did not report how they selected this architecture and parameter setting. For this reason we ran additional experiments using logistic regression as the discriminator model.

## 3.2 Datasets

We used the ZINC dataset (Irwin et al., 2012) provided in the GitHub repository associated with the original paper for some baselines and constrained improvement experiments.[4] The $J(m)$ scores are normalized with respect to this dataset (logP: mean 2.47, standard deviation 1.42, SAS: mean 3.05, standard deviation 0.831, ring penalty: mean 0.038, standard deviation 0.224).

## 3.3 Hyperparameters

Details about hyperparameter tuning were not provided in the original paper. Hyperparameters were set for all experiments in the code provided by the original reference and were re-used in this work. Furthermore, we performed some additional experiments in this work to better understand the influence of the $\beta$ parameter, model architecture, and training setup.

## 3.4 Experimental setup

The code used for this study is available on GitHub[5]. All experiments (including the failed attempts) were tracked using the Weights and Biases platform[6].

## 3.5 Computational requirements

The experiments do not necessarily need to be run on GPU for high performance, since only some of them use the neural network-based discriminator and since the model that is employed is small. The computational burden lies much more in the CPU-bound GA operations. Each GA generation takes $3\,\text{s}$ to $8\,\text{s}$ to run on one core (note that the original implementation already included some parallelization for the score calculation, and we used a population size of 500 for all experiments). Also the process memory requirements are low ($< 100\,\text{MB}$). It is practical to parallelize the score evaluations, especially for the Guacamol benchmark. Exact timings for all experiments and traces of hardware utilization can be found in the dashboard on the Weights and Biases platform.

# 4 Results and Discussion

Based on the code provided by the authors, we could reproduce results showing that a SELFIES-based GA can create molecules with high $J(m)$ (numerically higher than the references considered by Nigam et al. (2020)) and that the adaptive penalty can bias the generation of molecules that are similar to a reference set. We could also show that the SELFIES-based GA can outperform relevant baselines in some aspects but lacks intra-population diversity.

## 4.1 Baselines

Nigam et al. computed the penalized logP score for random SELFIES and found an impressively high score of $6.19 \pm 0.63$, outperforming many other generative techniques such as VAEs, (Gómez-Bombarelli et al., 2018; Dai et al., 2018) and Monte-Carlo Tree Search (MCTS) (Yang et al., 2017). We were able to reproduce the result when using the exact same version of the SELFIES library (v0.1.1). However, we obtained considerably lower scores when running this experiment using the latest version of the SELFIES library (v1.0.2), for which some bugs have been fixed and modifications such as kekulization as well as an extended alphabet have been added (i.e., more complex molecules can be assembled). To ensure compatibility of the latest SELFIES version with the old one, we ran experiments using the same alphabet (of 21 characters) as in the original study and an extended alphabet (of all 61 semantically robust characters), while keeping the constraints in both experiments to a maximum of 81 characters.

---

[4] The dataset is available as artifact on the Weights and Biases platform `https://wandb.ai/kjappelbaum/ga_replication_study/artifacts/raw_data/zinc_dearom/569e6cb67973e9983697/files`

[5] `https://anonymous.4open.science/r/041b3ae6-3bc2-4228-b56f-59e799f2c430/`

[6] `https://wandb.ai/kjappelbaum/ga_replication_study`

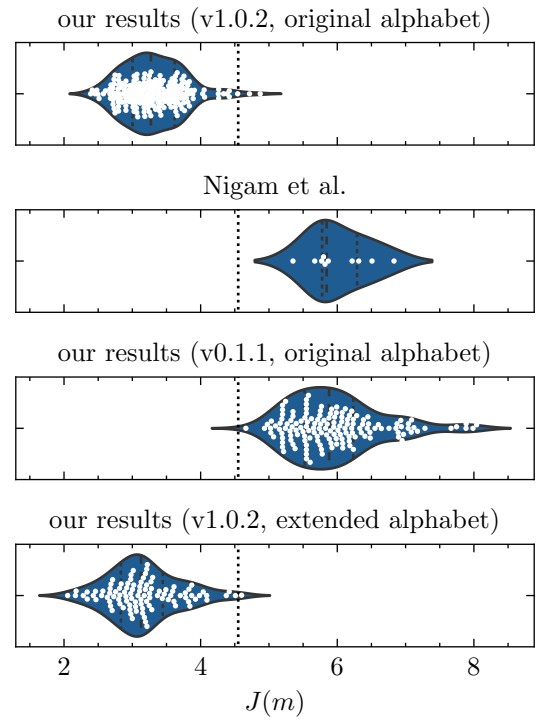

Figure 1: Violin plots of penalized logP ($J(m)$) distributions for different baseline experiments. Dashed lines in the violins indicate the quartiles. The dotted line indicates the best of dataset baseline (on the ZINC dataset used for this study), which is the most relevant baseline, as one wants to improve upon existing datasets with generative models.

The reasons for the high baseline score reported in the original paper is clear when one compares a few randomly sampled molecules: Using the original approach, the molecules have a higher density of aromatic rings, which increases the penalized logP score.

We suspect this high baseline is achieved due to an inductive bias introduced by the code, as we have found that some SELFIES created with the 21 character alphabet could not be decoded into valid SMILES using SELFIES version 0.1.1. We could decode those SELFIES using v1.0.2. and found $J(m) = -4.30 \pm 2.73$ (averaged over 50 000 SELFIES that raised errors in v0.1.1. but could be successfully decoded into SMILES without RDKit warning/error with v.1.0.2.) For this reason, we used version 1.0.2 of the code for all subsequent experiments. More importantly, our results show that choosing a relevant baseline is nontrivial as, for example, the choice of the alphabet can be thought of as inductive bias. Hence, following Brown et al. (2019), we suggest that the "best of dataset" is a more relevant baseline as any good generative model should outperform the training set or starting population. Furthermore, we find that comparing algorithm performance based on penalized logP scores without setting a limit to molecular size can be misleading, as this score can be maximized by simply increasing the molecular size, whereas from a practical point of view such large molecules have limited potential as a drug molecule (for example, very large logP values can lead to pharmakokinetic problems). We note that Nigam et al. (2020) did apply a relatively high threshold limit to the maximum molecular size (81 characters). In contrast, Jensen (2019) applied a threshold of $39.15 \pm 3.50$ non hydrogen atoms for their GA.

## 4.2   SELFIES GA without discriminator (Claim 1)

Using the GA, Nigam et al. (2020) could achieve nearly double the highest reported penalized logP score reported so far by evolving SELFIES seeded with a methane molecule. As noted earlier, Nigam et al. applied a higher threshold on the maximum number of heavy atoms than some other previous works. We find a comparable score of $J(m) = 11.911 \pm 1.262$ (averaged over 10 runs) using the same approach as in the original paper, albeit the highest-scoring molecules have a heavy atom count of $58.800 \pm 9.775$.

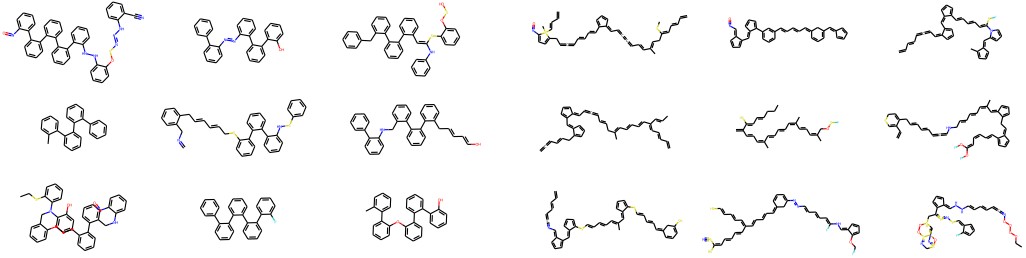

(a) Random sample of molecules created in the base-line experiment by the original authors.

(b) Random sample of molecules created in our base-line experiment.

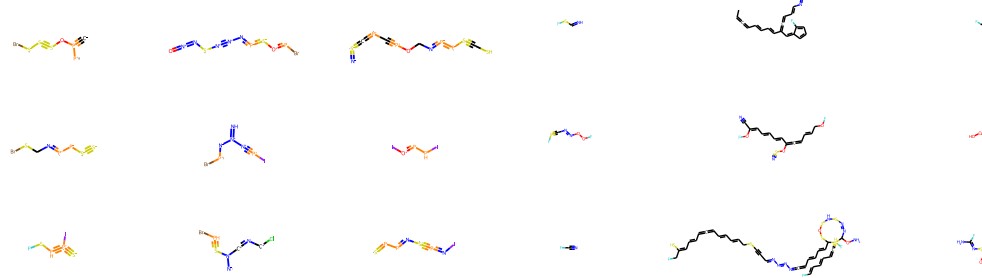

(c) Random sample of molecules created in our base-line experiment with extended alphabet.

(d) Random sample of molecules that could not be decoded to valid SMILES using SELFIES v0.1.1, but could be decoded using SELFIES v1.0.2.

Figure 2: Randomly sampled molecules for our different baseline approaches.

### 4.3 SELFIES GA with discriminator (Claim 2)

In the original paper, the discriminator neural network is trained jointly on mutated SELFIES strings, and data from the reference as a binary classifier, where molecules from the reference set are labeled with 1. Increasing $\beta$ hence means biasing the GA to create molecules that are similar to the reference set. This also means that molecules that survive for longer times will tend to reduce $D(m)$. The intuition behind this is that a large discriminator term will force the chemistry to be similar to the reference set and cause long-surviving families to die. Note that in this study we also investigated negative $\beta$, which effectively penalizes molecules that are similar to the reference database.

Similar to Nigam et al. we find that large $\beta$ yields $J(m)$ around zero (Figure 3), as the scores are normalized with respect to the reference set. From Figure 4, we can clearly see that if we reward similarity to the reference set (large $\beta$), the generated molecules are more complex and contain a larger variety of functionalities. It also interesting to use large, negative $\beta$. This effectively penalizes similarity to the reference set. For $\beta = -100$ we observe that the GA generates small molecules, e.g. nitrous oxide, that often do not contain any carbon.

Our GA runs outperform the ones reported in the original paper in terms of $J(m)$, but also in our case, the GA exploits deficiencies in the scoring function.

#### 4.3.1 Internal diversity quantification

Nigam et al. emphasize the importance of diversity in the generative molecules in qualitative terms. It is instructive to *quantify* how the different generative approaches discussed in this work influence the diversity. To do this, we follow the approach proposed by Benhenda (2017) and calculate the mean pairwise Tanimoto distance $T_d(x, y)$ of the Morgan fingerprints (Rogers and Hahn, 2010) of the set $\mathcal{A}$ best-performing molecules obtained in the last generation, that is

$$\text{internal diversity} = \sum_{(x,y)\in\mathcal{A}\times\mathcal{A}} T_d(x,y)/|\mathcal{A}|^2. \tag{2}$$

From Figure 5 we see that a low $\beta$ increases the similarity between subsequent best performing molecules within a generation, whereas high, positive $\beta$ decreases the similarity between the highest-scoring molecules—but also forces $J(m)$ to be near the mean of the reference set (Figure 3).

For some applications it is important to generate a diverse set of high performing molecules, for example, as in the case of drug discovery where the chances of commercialization of an early lead structure are quite low. For this reason, we

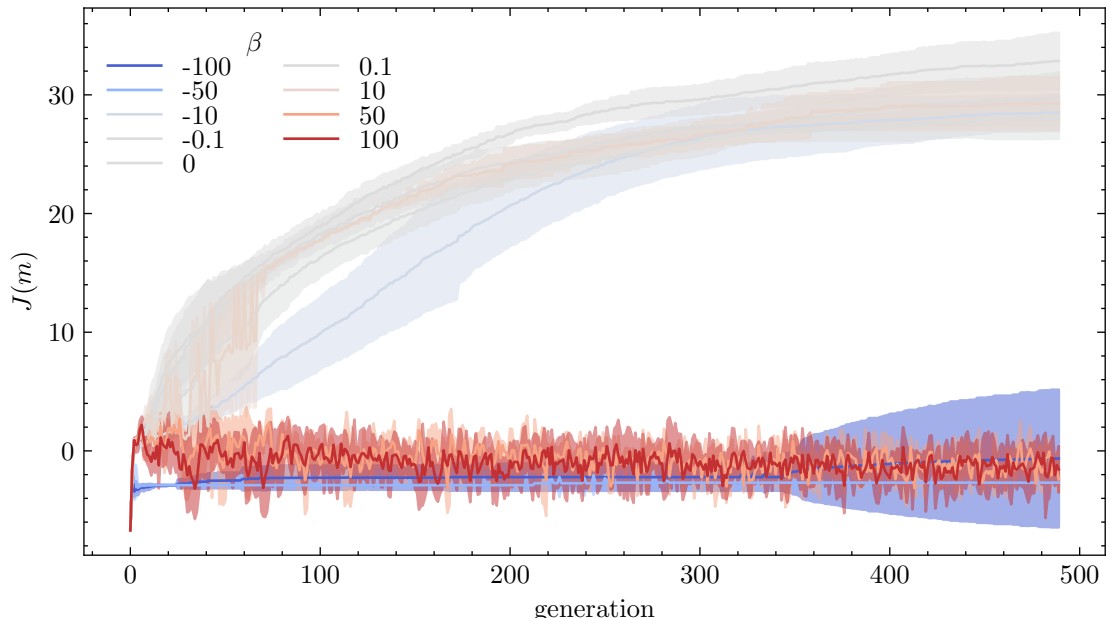

Figure 3: $J(m)$ evolution as a function of $\beta$. The means and the $1\sigma$ interval of multiple runs are shown by the solid lines and shaded regions, respectively.

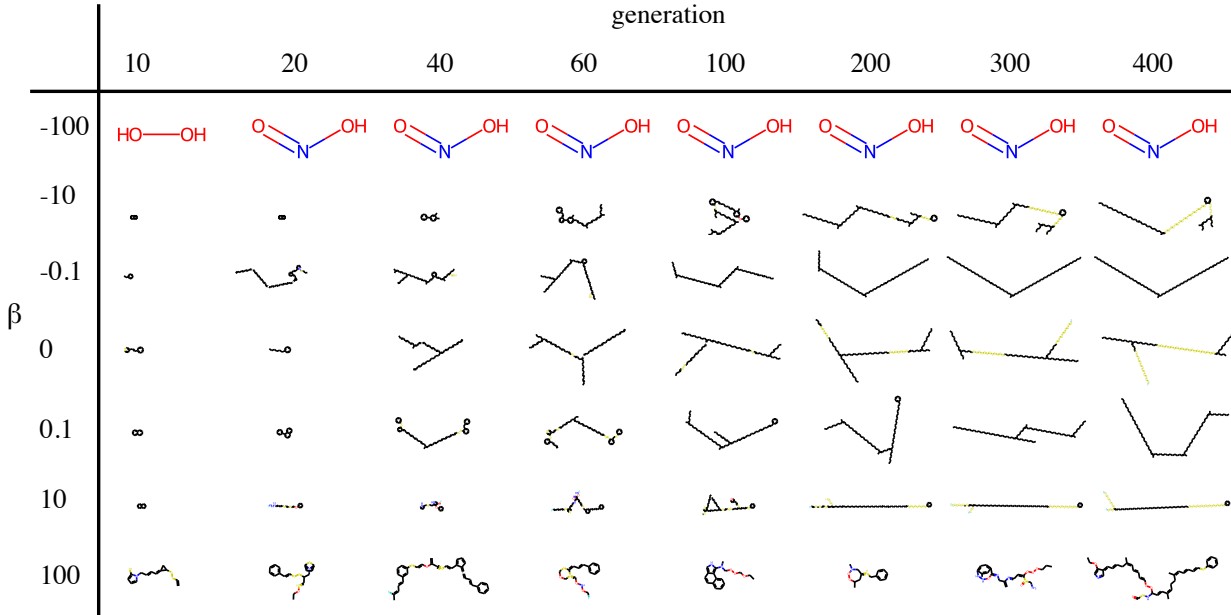

Figure 4: Evolution of the best performing structures generated by the GA for different $\beta$. The means and the $1\sigma$ interval of multiple runs are shown by the solid lines and shaded regions, respectively.

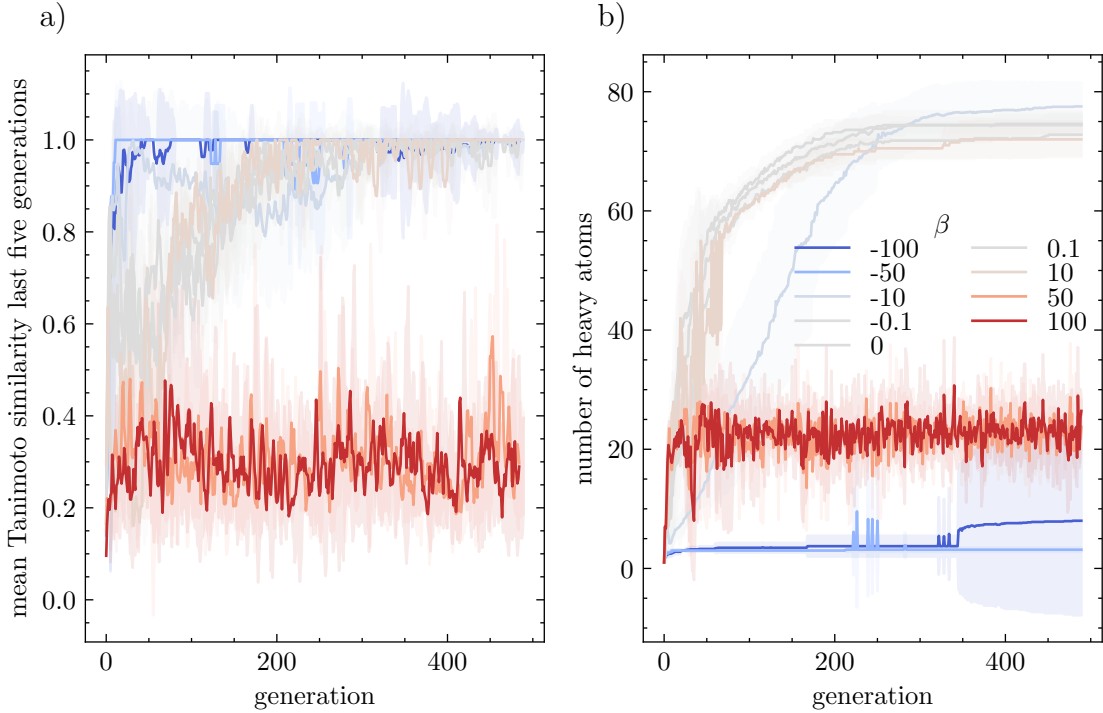

Figure 5: a) Evolution of internal similarity as a function of $\beta$. Mean pairwise Tanimoto similarity of the Morgan fingerprints of the best performing molecules in the last five generations. In general, the GA converges to the creation of similar molecules except for high $\beta > 10$. b) Evolution of the heavy atom count as a function of $\beta$. For high $\beta > 10$, the generated molecules are significantly shorter than those for low $\beta$. The means and the $1\sigma$ interval are shown by the solid lines and shaded regions, respectively.

analyzed how many unique and diverse molecules the GA generates. From Figure 6 we see that the fraction of unique molecules and the diversity within a generation is decreasing as the optimization progresses. For the fraction of unique molecules, we find that all positive and small negative values of beta $\beta$ show similar behaviors, yielding about $40\%$ unique molecules after 500 generations. Only the lowest values of $\beta$ do not reach more than $20\%$ unique molecules. In terms of intra-population diversity, we find that high values of $\beta$ biases the generation to more diverse populations.

## 4.4 Time-adaptive penalty (Claim 2)

A typical issue with GAs is that they can get stuck and only return the same molecule for multiple iterations. To mitigate this problem, Nigam et al. proposed a time-adaptive penalty in which they added a $1000D(m)$ term to the fitness function only if the maximum fitness stagnated at the *exact same maximum fitness value* for the last five generations. The authors could significantly increase the maximum $J(m)$ found by the GA using this approach but did not justify their choice of $\beta$. We also performed this experiment with different $\beta$ and quantified the evolution of internal diversity and molecular sizes. From Figure 7 we find that the time-adaptive penalty frequently reduces the $J(m)$ and mean Tanimoto similarity, after which both quickly recover. This, on average, reduces the mean Tanimoto similarity of the best performing molecules with increasing $\beta$. Notably, this average is higher than for the case in which the same $\beta$ is applied all the time. By analyzing multiple independent runs we can quantify how the generator stagnates under different $\beta$. Stagnation is defined here in the same way Nigam et al. defines it in their code, i.e., if maximum $J(m)$ has not changed for five generations. From Figure 8, we see that runs at low $\beta$ have a higher chance of getting stuck for a substantial number of generations due to the penalty not having a large enough effect on the fitness function. In one case ($\beta = 200$), the generator stagnated for $80\%$ of generations.

## 4.5 Constrained optimization

A common benchmark task for generative models is to generate molecules that are similar to some target. Nigam et al. adapted the experimental framework from You et al. (2018) in which one aims to improve the $J(m)$ for 800 low performing structures from the references set. Similar to Nigam et al. we performed this experiment with two similarity

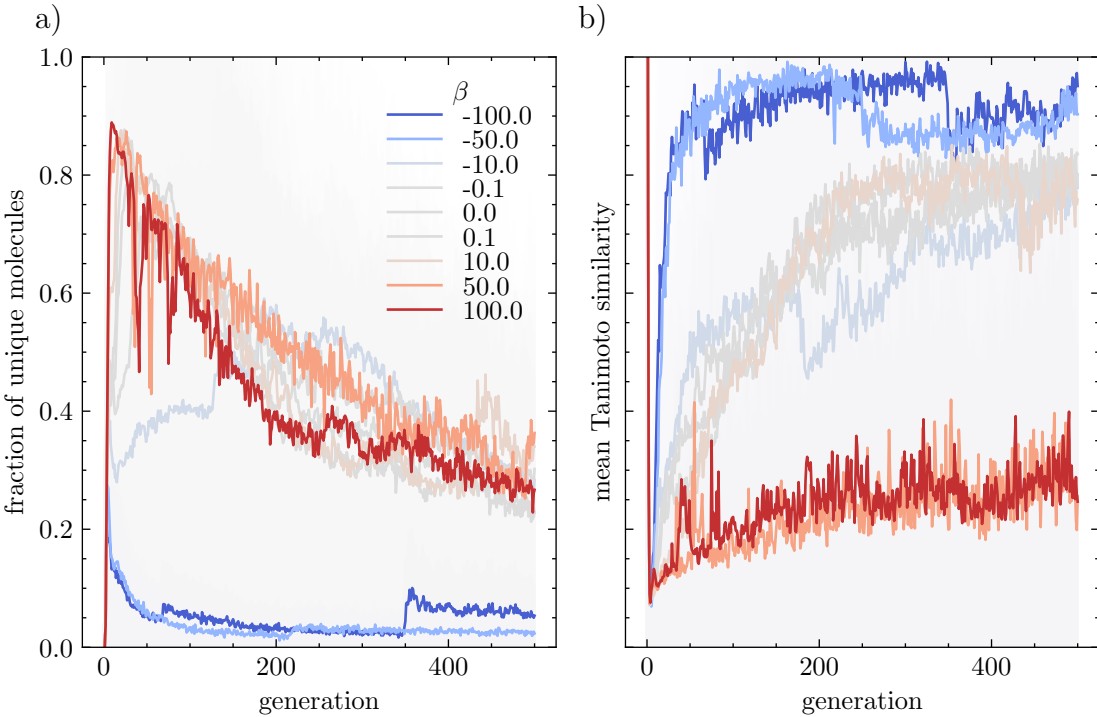

Figure 6: a) Fraction of unique molecules per generation for different $\beta$. b) Internal diversity (estimated using the mean Tanimoto distance of radius 2 Morgan fingerprints) of the population per generation for different $\beta$. To make the estimation of the internal diversity more efficient we calculate it on a random subset of 100 out of 500 molecules from the population.

thresholds $\delta = \{0.4, 0.6\}$ between the Morgan fingerprints of the target molecule and the GA output, but limited our study to 285 molecules for $\delta = 0.4$ and 120 for $\delta = 0.6$. Overall, our findings agree with the ones of Nigam et al., even though we observe slightly lower success rates (we count a run as success when the improvement in the score $> 0$). More specifically, we fail to improve $J(m)$ for more molecules than Nigam et al. and could not achieve the perfect success rate of $100\,\%$. For a similarity threshold of $\delta = 0.4$, we find an average improvement of $4.11 \pm 1.58$ and a success rate of $99\,\%$. For the larger tolerance of $\delta = 0.6$, we find an average improvement of $1.87 \pm 1.32$ and a success rate of $100\,\%$. We find that the mean improvements agree to within the error margins of those reported by Nigam et al. (2020).

## 4.6 Optimization of multiple properties

To demonstrate the versatility of their approach, Nigam et al. also incorporated the quantitative estimate of drug-likeness (QED) score (Bickerton et al., 2012) into their loss function. This is an interesting problem as solubility indicator, logP, and the QED score cannot be maximized at the same time, i.e., one would like to recover the Pareto front. From Figure 9a) we see that the best performing molecules in a generation quickly converge to one point of the Pareto front, i.e., after 100 generations we do not observe much evolution of the properties of the top scoring molecules. From Figure 9b) we find that the molecules in the final generation have a good Pareto front coverage, but only in few instances outperform the original Pareto optimal molecules from the ZINC dataset (we find the hypervolume of the Pareto front of the ZINC database to be 7.37 and the one found by the GA to be 7.14 w.r.t. the nadir point of the ZINC dataset).

## 4.7 Similarity-triggered adaptive penalty

Nigam et al. used the time-adaptive penalty to encourage explorative behavior. They measure estimated stagnation based on constant maximum fitness. We anticipated that better results with greater molecular diversity could be achieved by using a penalty that is applied based on an explicit structure similarity criterion. Therefore, we implemented an adaptive penalty based on the internal molecular similarity of the best-performing molecules in the last five generations (measured in terms of the average pairwise distances of the Morgan fingerprints (Rogers and Hahn, 2010) as proposed

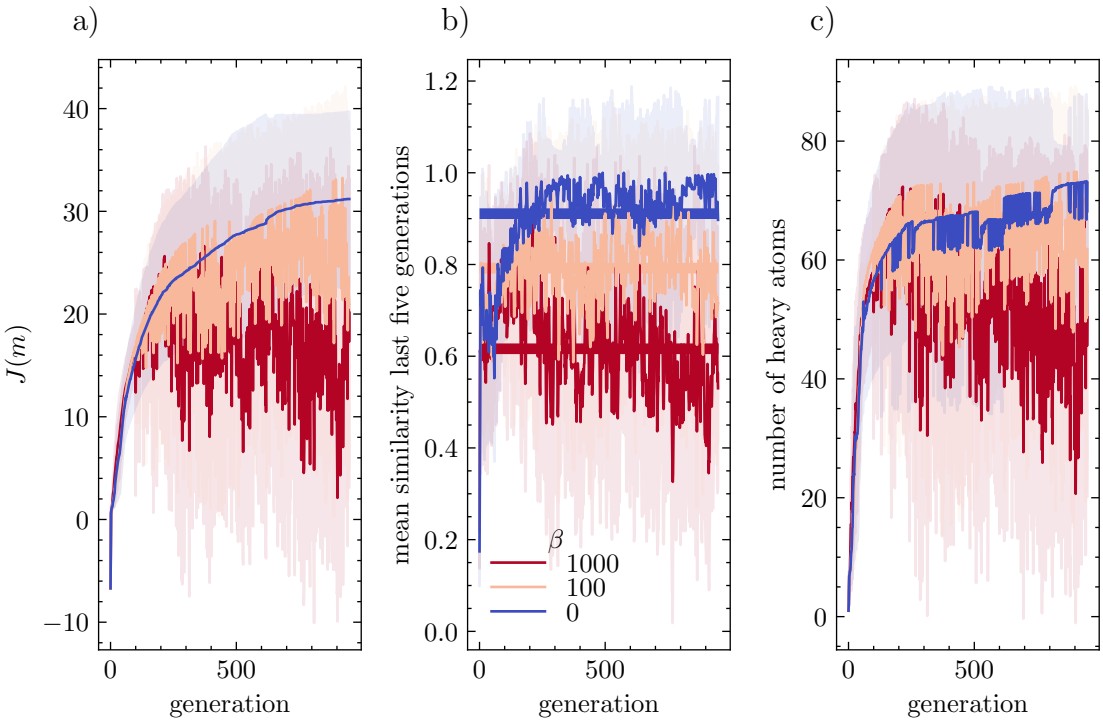

Figure 7: Evolution of $J(m)$, similarity, and molecular size under a GA with the time-adaptive penalty proposed by Nigam et al. Thick horizontal lines in panel b) indicate the mean Tanimoto similarity after generation 100. The means and the $1\sigma$ interval of multiple runs are shown by the solid lines and shaded regions, respectively. Note that the standard deviations between runs increase with increasing $\beta$.

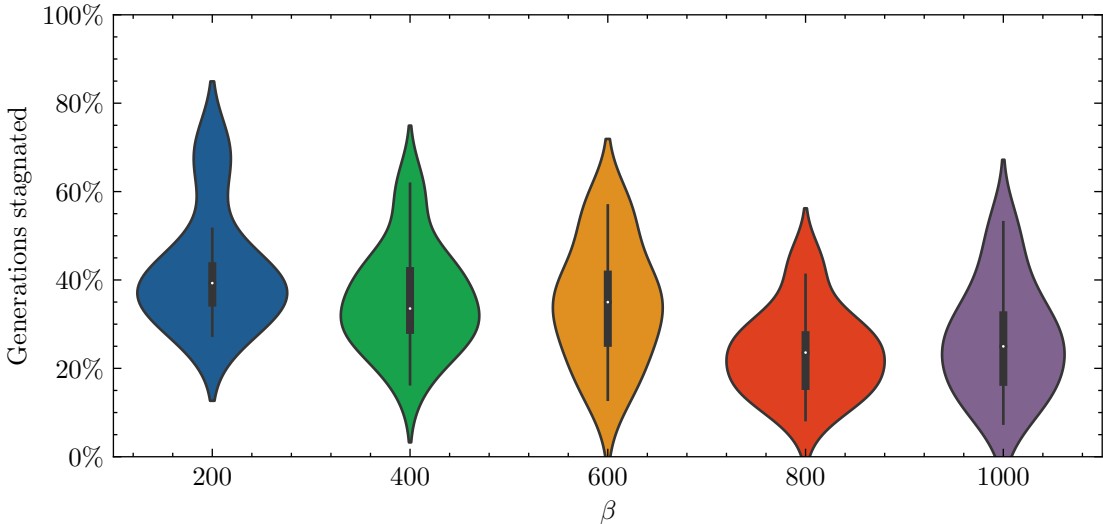

Figure 8: Comparison of generator stagnation over 18 independent runs for multiple time adaptive penalty parameter ($\beta$). Stagnation is defined here in the same way Nigam et al. defines it in their code, i.e. if maximum $J(m)$ has not changed for five generations.

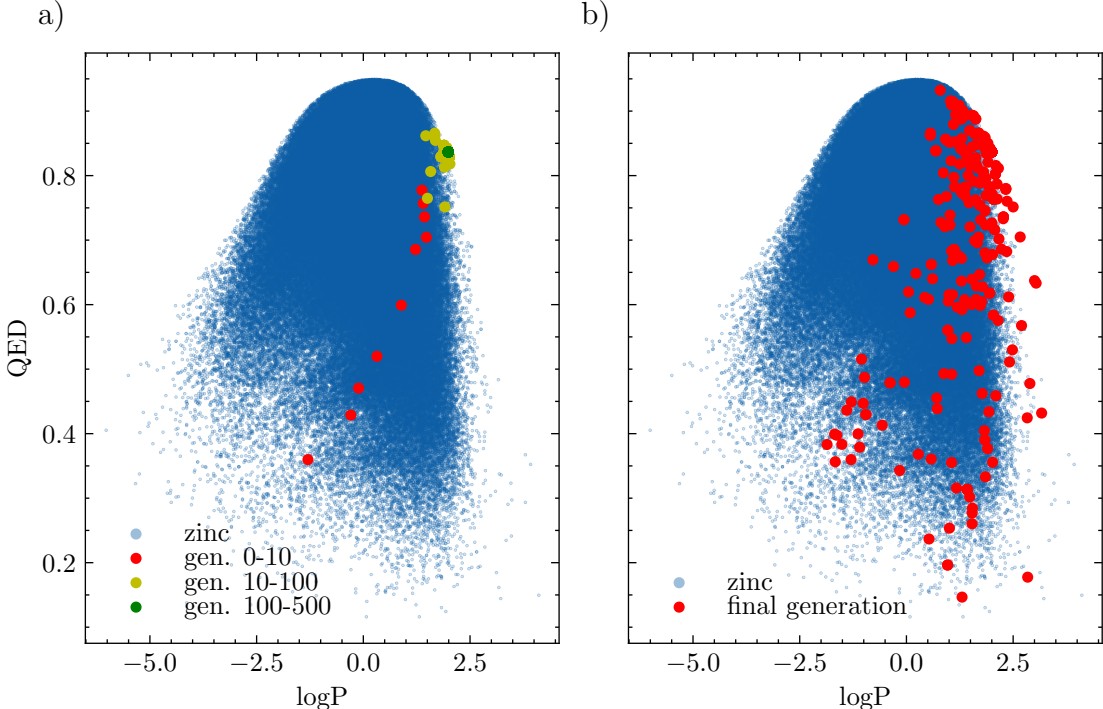

Figure 9: a) Evolution of QED and penalized logP for the best-performing molecules in a generation. b) Distribution of penalized logP and QED for the molecules of the final generation compared to the distribution in the ZINC dataset.

by Benhenda (2017)). In contrast to Nigam et al., who started to impose this penalty after 100 iterations, we started imposing this adaptive penalty after 20 generations. We did this based on the consideration that after 20 generations, the GA should have enough time to evolve away from the methane starting point to some decently performing molecule (see for example Figure 9a); note that we did not tune this hyperparameter.

From Figure 10 we see that the success with this penalty highly depends on the similarity threshold. If it is chosen too small (i.e., close to non-significant following Vogt and Bajorath (2020)) the fitness does not improve. We find that by using the similarity-triggered penalty we can keep the similarity between subsequent top-performing molecules under a user-defined threshold. Interestingly, we also observe that this does not necessarily compromise $J(m)$. We find scores comparable to the highest ones shown in Figure 3—however, we do not converge to a similarity of close to unity using the similarity-triggered penalty.

### 4.8 Influence of the labeling convention

In the setup proposed by Nigam et al. long-surviving molecules will return low discriminator scores $D(m)$. One extension of this approach can be to penalize long-surviving molecules, i.e., subtract a penalty from the fitness for molecule classes that are long-surviving. This can be achieved by reversing the way in which the discriminator model is trained, i.e., predicting a score $> 0.5$ for long-surviving molecules (instead of 0 in the original approach). Figure 11 shows some results under this framework. We see that with the reversed labeling convention, $J(m)$ is not as sensitive to changes to $\beta$ than compared with the labeling convention chosen by Nigam et al. (2020). This is also consistent with what we observe in the evolution of the discriminator scores (Figure 12). The reversed labels fail to reach this objective (for large $\beta$ in the original labeling convention we can bias the GA to the generation of molecules with large $D(m)$). We suspect that this behavior occurs, because it is easier for the model to learn the similarity to the reference set than it is to learn the stagnation, potentially due to low intra-population diversity. These observations might imply that the discriminator term rather works via biasing to similarity to a reference set rather than by penalizing long-surviving molecules.

To understand this behavior better, we compared the evolution of the best performing molecules for different $\beta$. We find that the $D(m)$ penalty encourages exploratory behavior in the first few generations. After 100–200 generations, the GA begins to recognize patterns (long chains with sulfur) that it can easily exploit and outperforms $J(m)$ that have been

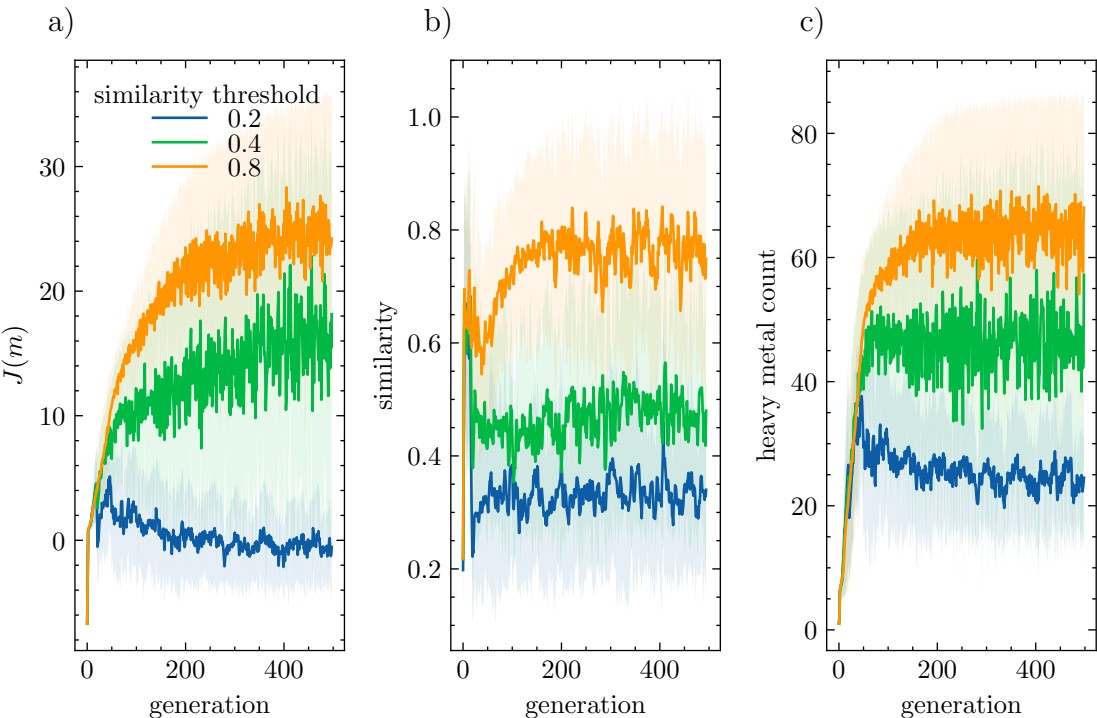

Figure 10: Evolution of properties using similarity-triggered discriminator penalty. Solid lines indicate means of independent runs, shaded regions indicate $1\sigma$ regions.

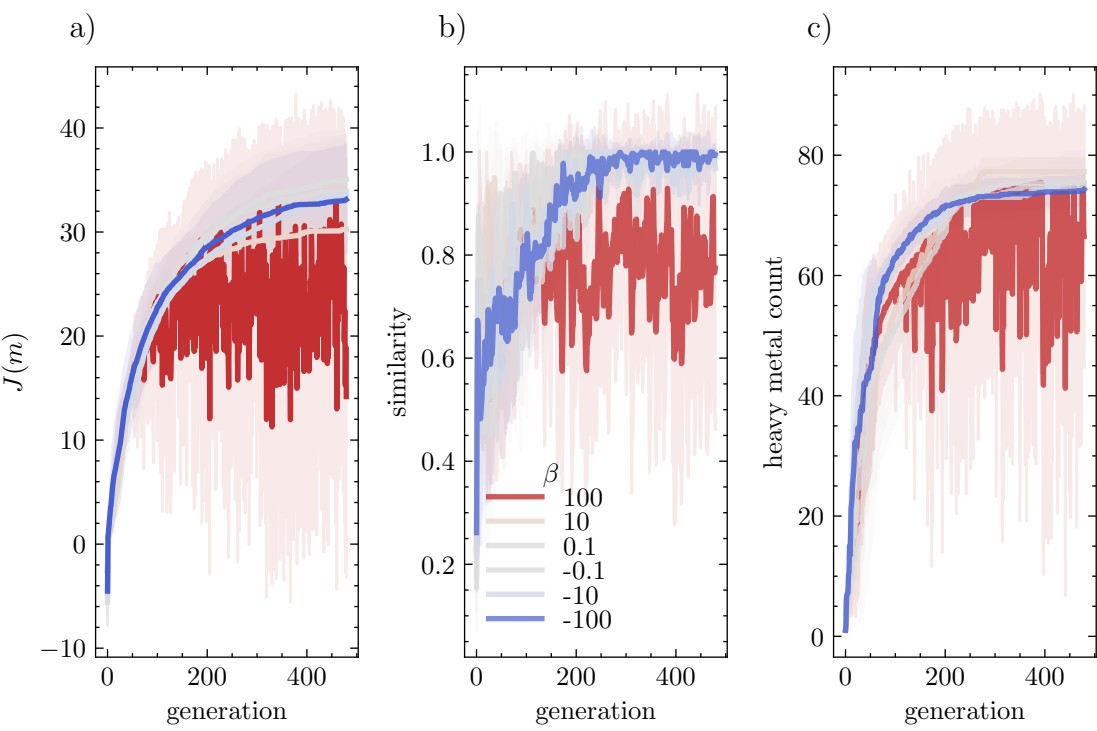

Figure 11: Evolution of properties under GA with discriminator trained with reversed labels. The means and the $1\sigma$ interval of multiple runs are shown by the solid lines and shaded regions, respectively.

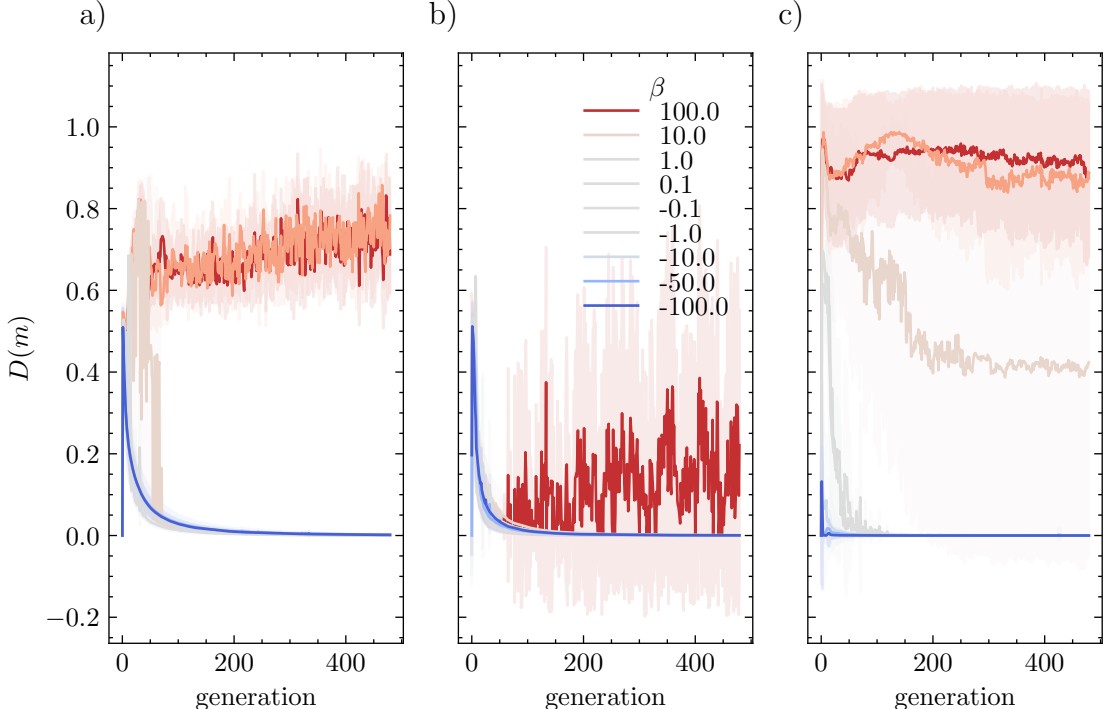

Figure 12: Evolution of discriminator scores $D(m)$ with continuously applied $\beta D(m)$ term under the original labeling convention (a) and flipped labeling convention (b). The means and the $1\sigma$ interval of multiple runs are shown by the solid lines and shaded regions, respectively.

previously reported. Again we note, however, that this is mainly attributed to the larger molecular size (see Figure 11c). For large $\beta$ (i.e., encouraging similarity) we find that the scaffold mainly evolves by elongating linear chains.

In comparison to Figure 4, we also observe that molecules do not contain functionalities as complex as those found for large $\beta$ with the original labeling, with which we can explicitly reward similarity to the reference set. Note that with the flipped labels, the molecules from the reference set will yield a discriminator score of $D(m) = 0$.

## 4.9 Influence of discriminator model architecture

Nigam et al. did not rationalize the choice of the model architecture. To understand the sensitivity of the algorithm performance to the architecture of the discriminator model, we replaced the neural network with a logistic regression model (i.e., we use only one hidden layer) and ran experiments with different $\beta$. From Figure 14, we see that the simpler model can also be used successfully as a discriminator and reproduce the same dependence of $J(m)$ on $\beta$ as seen in the runs with neural network-based discriminator model. Analysis of the evolution reveals an interesting difference compared to the deeper model—the logistic regression performs worse in penalizing long-surviving molecules. In other words, we observe higher similarities of consecutive molecules for high $\beta$.

## 4.10 Performance on the GuacaMol benchmark

As a subsequent benchmark study, we tested the GA on the goal-directed benchmarks from the GuacaMol library, which have been shown to be more challenging than the optimization of simple molecular properties such as $J(m)$ (Brown et al., 2019). Furthermore, to test whether the SMILES-based GA (Yoshikawa et al., 2018) can improve over the "best of dataset" within the GuacaMol benchmark suite, we initialized the GA with the best scoring molecules from the GuacaMol dataset. Note that we did not perform an exhaustive and systematic hyperparameter optimization[7], wherefore the scores (Table 1) represent a lower limit on the performance of the SELFIES-GA.

---

[7]we tested 10, 100, and 400 generations ($n$), and different $\beta \in \{0, 100, 1000\}$, as well as similarity thresholds ($\delta \in \{0.2, 0.4, 0.8\}$), but did not try the extended alphabet in the new SELFIES version, different population sizes, different patience of the time-adaptive penalty, or multiple random restarts. We only considered the molecules in the *final generation* for the scoring.

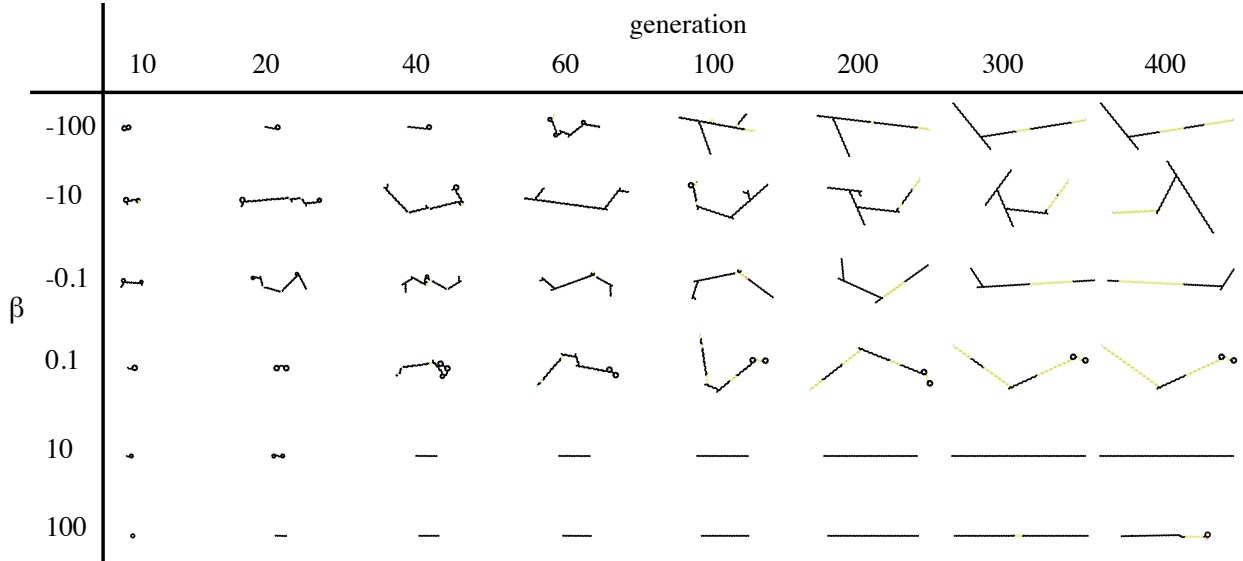

Figure 13: Evolution of the top-scoring structures for different $\beta$ with the $D(m)$ trained with labels flipped w.r.t. the original implementation.

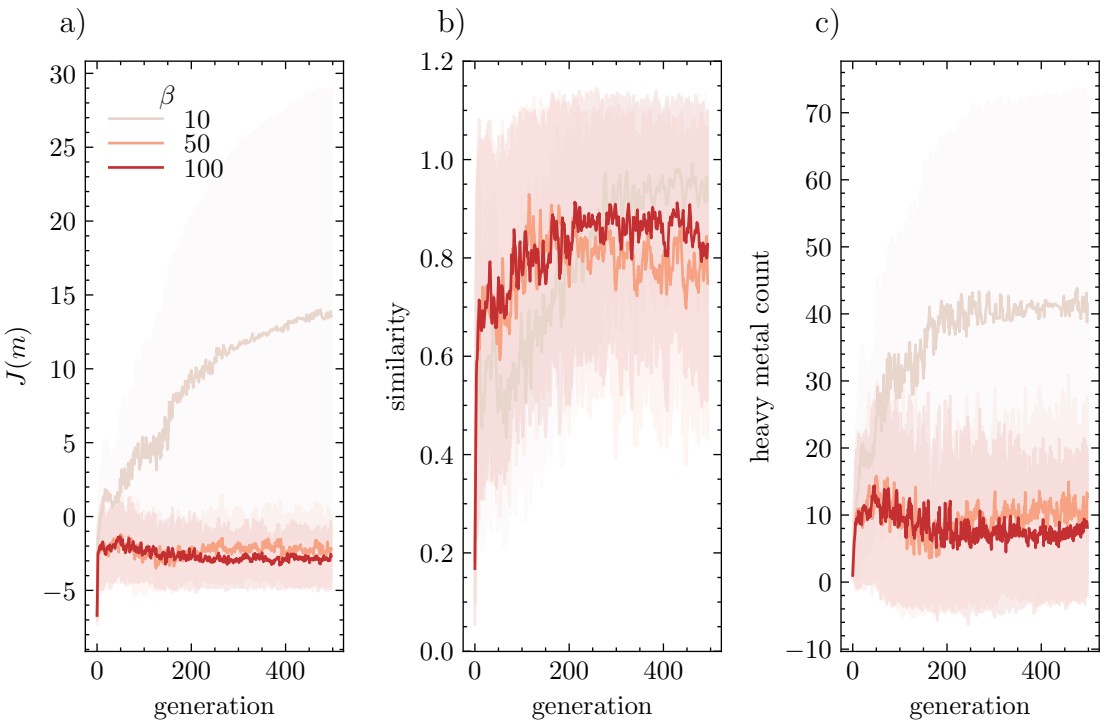

Figure 14: Comparison of the GA performance with logistic regression as discriminator and a multilayer neural network. The means and the $1\sigma$ interval of multiple runs are shown by the solid lines and shaded regions, respectively.

We did not find the SELFIE-bases GA to outperform the best performances in the leaderboard, but—even without systematic parameter tuning—we did find it to mostly improve beyond the "best of dataset" benchmark, and in some cases (e.g., Celecoxib rediscovery), also to perform better than the grammatical evolution of SMILES reported by Yoshikawa et al. (2018) (SMILES-GA baseline). Most striking are the lower scores on the isomer discovery tasks ($C_{11}H_{24}$, $C_9H_{10}N_2O_2PF_2Cl$). Even with discriminator-based penalty the GA returns only few unique molecules, which reflects the findings shown in Figure 6 in which the GA converges to a low number of unique molecules. In the case of isomer discovery, this issue is exacerbated since all valid isomers will have the same fitness, and hence the same probability of being replaced with mutations on one of the top-scoring molecules. Additionally, there is no term that encourages diversity within a population. For these reasons, we observe that the isomer discovery scores improve if we run the GA for fewer iterations (we see from Figure 6 that the fraction of unique molecules decreases over the course of the optimization). On the other hand, if only top-1 scores are considered (as in the rediscovery tasks) the SELFIES-GA outperforms the SMILES-GA baseline. For these tasks, we see that running the GA for more iterations improves the scores.

## 5 Conclusions and Future Work

In summary, we could easily reproduce high performance in the optimization of the penalized logP score using the SELFIES based GA. We could also reproduce behavior showing that an adaptive penalty can be used to bias the GA to the generation of molecules that are similar to a reference database. Building on top of the work of Nigam et al., we also *quantified* the evolution of similarity for different $\beta$ and investigated the influence of the discriminator model, the labeling convention, and proposed a new way of applying the time-adaptive penalty, which we found to outperform the original results.

However, our analysis also highlights the need for fairer comparisons when benchmarking generative algorithms. The GA mostly outperforms the scores reported so far mainly by exploiting deficiencies of the $J(m)$ scoring function, which results in the generation of long molecules with unfeasible substructures (e.g., sulfur chains). For this reason, we also benchmarked the performance of the algorithm on the GuacaMol set of benchmarks. Clearly, such benchmarks allow for a much more holistic and fairer comparison of algorithms than the use of the penalized logP score. The issue with the latter is not only that it is easy to exploit, but that authors use different thresholds on the maximum number of atoms or different character sets which can make it more challenging to reproduce results.

Notably, we observed that there is often a relatively low diversity in the molecules within a generation—which leads to low scores on some of the GuacaMol benchmarks (in which not only the top-1 score is considered). Since a large pool of diverse molecules is quite relevant for applications with high failure rates (e.g., drug discovery), we propose the addition of penalty terms of the form $1 - \text{internal population diversity}$ to the fitness so that the GA is biased toward a higher intra-generation diversity.

## Acknowledgments

Our study made use of the following libraries: ccbmlib (Vogt and Bajorath, 2020), click (Ronacher, 2021), GuacaMol (Brown et al., 2019), jupyter (Kluyver et al., 2016), matplotlib (Hunter, 2007), numpy (Harris et al., 2020), pandas (McKinney, 2010), pyepal (Jablonka et al., 2020), pytorch (Paszke et al., 2019), RDKit (Landrum et al.), SciencePlots (Garrett, 2020), scikit-learn (Pedregosa et al., 2011), scipy (SciPy 1.0 Contributors et al., 2020), SELFIES (Krenn et al., 2020).

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

---

Hence, if the GA fails to generate many unique molecules this will lower the score. We chose this implementation for a fairer comparison with the SMILES-GA baseline.

Table 1: Scores on the GuacaMol v2 benchmarks (Brown et al., 2019). MPO objectives are multi-property objectives, and Hop benchmarks aim to maximize the similarity to a target while keeping or excluding specific functionalities. Median benchmarks optimize the similarity to multiple molecules at once. For all runs, we used a generation size of 500, the alphabet employed in the original paper, and seeded the first population with the best scoring molecules from the reference dataset provided by the GuacaMol library. "Leading" refers to the best score reported in the leaderboard (`https://www.benevolent.com/guacamol`).

| benchmark | SMILES GA | leading | original $\beta = 1000$, $n = 100$ | $\beta = 1000$, $\delta = 0.4$, $n = 10$ | $\beta = 1000$, $\delta = 0.8$, $n = 100$ | $\beta = 1000$, $\delta = 0.2$, $n = 100$ | $\beta = 100$, $\delta = 0.8$, $n = 100$ | $\beta = 100$, $\delta = 0.8$, $n = 400$ | $\beta = 100$, $\delta = 0.2$, $n = 400$ |
|---|---|---|---|---|---|---|---|---|---|
| $C_{11}H_{24}$ | 0.83 | 0.99 | 0.01 | **0.30** | 0.01 | 0.01 | 0.03 | 0.01 | 0.01 |
| $C_9H_{10}N_2O_2PF_2Cl$ | 0.89 | 0.98 | 0.02 | **0.56** | 0.02 | 0.16 | 0.06 | 0.00 | 0.01 |
| Osimertinib MPO | 0.89 | 0.95 | 0.39 | **0.78** | 0.65 | 0.43 | 0.33 | 0.33 | 0.37 |
| Fexofenadine MPO | 0.93 | 1.00 | 0.36 | **0.73** | 0.46 | 0.52 | 0.39 | 0.35 | 0.40 |
| Ranolazine MPO | 0.88 | 0.92 | 0.41 | **0.74** | 0.34 | 0.35 | 0.37 | 0.34 | 0.35 |
| Celecoxib rediscovery | 0.73 | 1.00 | 0.81 | 0.58 | 0.78 | 0.78 | 0.78 | 0.84 | **1.00** |
| Troglitazone rediscovery | 0.51 | 1.00 | 0.64 | 0.46 | 0.57 | **0.70** | 0.65 | 0.57 | 0.64 |
| Thiothixene rediscovery | 0.60 | 1.00 | 0.60 | 0.47 | 0.69 | 0.57 | 0.70 | 0.57 | **0.74** |
| Aripiprazole similarity | 0.83 | 1.00 | 0.40 | **0.58** | 0.29 | 0.32 | 0.33 | 0.41 | 0.44 |
| Albuterol similarity | 0.91 | 1.00 | **0.71** | 0.66 | 0.41 | 0.41 | 0.59 | 0.41 | 0.52 |
| Mestranol similarity | 0.79 | 1.00 | 0.35 | **0.57** | 0.38 | 0.35 | 0.33 | 0.29 | 0.29 |
| Median molecules 1 | 0.33 | 0.44 | 0.13 | **0.27** | 0.15 | 0.15 | 0.15 | 0.15 | 0.15 |
| Median molecules 2 | 0.38 | 0.43 | 0.13 | 0.25 | **0.27** | 0.17 | 0.14 | 0.13 | 0.16 |
| Perindopril MPO | 0.66 | 0.81 | 0.51 | **0.55** | 0.49 | 0.31 | 0.30 | 0.32 | 0.27 |
| Amlodipine MPO | 0.72 | 0.89 | 0.48 | **0.62** | 0.60 | 0.28 | 0.56 | 0.28 | 0.35 |
| Sitagliptin MPO | 0.69 | 0.89 | 0.32 | **0.51** | 0.31 | 0.32 | 0.35 | 0.31 | 0.33 |
| Zaleplon MPO | 0.41 | 0.75 | 0.24 | **0.50** | 0.27 | 0.27 | 0.25 | 0.25 | 0.25 |
| Valsartan SMARTS | 0.55 | 0.99 | 0.36 | **0.69** | 0.36 | 0.36 | 0.36 | 0.36 | 0.36 |
| Scaffold Hop | 0.97 | 1.00 | 0.45 | **0.66** | 0.61 | 0.59 | 0.66 | 0.39 | 0.65 |
| Deco Hop | 0.88 | 1.00 | 0.55 | 0.81 | 0.46 | **0.86** | 0.71 | 0.43 | 0.44 |

N. Brown, M. Fiscato, M. H. Segler, and A. C. Vaucher. GuacaMol: Benchmarking models for de novo molecular design. *Journal of Chemical Information and Modeling*, 59(3):1096–1108, Mar. 2019. doi: 10.1021/acs.jcim.8b00839. URL https://doi.org/10.1021/acs.jcim.8b00839.

H. Dai, Y. Tian, B. Dai, S. Skiena, and L. Song. Syntax-directed variational autoencoder for structured data. *CoRR*, abs/1802.08786, 2018. URL http://arxiv.org/abs/1802.08786.

N. De Cao and T. Kipf. Molgan: An implicit generative model for small molecular graphs. *arXiv:1805.11973*, 2018.

J. Devillers. *Genetic algorithms in molecular modeling*. Academic Press, London San Diego, 1996. ISBN 9780080532387.

P. Ertl and A. Schuffenhauer. Estimation of synthetic accessibility score of drug-like molecules based on molecular complexity and fragment contributions. *Journal of Cheminformatics*, 1(1), June 2009. doi: 10.1186/1758-2946-1-8. URL https://doi.org/10.1186/1758-2946-1-8.

J. D. Garrett. SciencePlots (v1.0.6), Oct. 2020. URL http://doi.org/10.5281/zenodo.4106650.

R. Gómez-Bombarelli, J. N. Wei, D. Duvenaud, J. M. Hernández-Lobato, B. Sánchez-Lengeling, D. Sheberla, J. Aguilera-Iparraguirre, T. D. Hirzel, R. P. Adams, and A. Aspuru-Guzik. Automatic chemical design using a data-driven continuous representation of molecules. *ACS Central Science*, 4(2):268–276, Jan. 2018. doi: 10.1021/acscentsci.7b00572. URL https://doi.org/10.1021/acscentsci.7b00572.

I. J. Goodfellow, J. Pouget-Abadie, M. Mirza, B. Xu, D. Warde-Farley, S. Ozair, A. Courville, and Y. Bengio. Generative adversarial networks. *1406.2661*, 2014.

C. R. Harris, K. J. Millman, S. J. van der Walt, R. Gommers, P. Virtanen, D. Cournapeau, E. Wieser, J. Taylor, S. Berg, N. J. Smith, R. Kern, M. Picus, S. Hoyer, M. H. van Kerkwijk, M. Brett, A. Haldane, J. F. del Río, M. Wiebe, P. Peterson, P. Gérard-Marchant, K. Sheppard, T. Reddy, W. Weckesser, H. Abbasi, C. Gohlke, and T. E. Oliphant. Array programming with NumPy. *Nature*, 585(7825):357–362, Sept. 2020. ISSN 0028-0836, 1476-4687. doi: 10.1038/s41586-020-2649-2.

J. D. Hunter. Matplotlib: A 2D Graphics Environment. *Comput. Sci. Eng.*, 9(3):90–95, 2007. ISSN 1521-9615. doi: 10.1109/MCSE.2007.55.

J. J. Irwin, T. Sterling, M. M. Mysinger, E. S. Bolstad, and R. G. Coleman. ZINC: A free tool to discover chemistry for biology. *Journal of Chemical Information and Modeling*, 52(7):1757–1768, June 2012. doi: 10.1021/ci3001277. URL https://doi.org/10.1021/ci3001277.

K. M. Jablonka, G. M. Jothiappan, S. Wang, B. Smit, and B. Yoo. Bias free multiobjective active learning for materials design and discovery. *Chemrxiv preprint*, Nov. 2020. doi: 10.26434/chemrxiv.13200197.v1. URL https://doi.org/10.26434/chemrxiv.13200197.v1.

J. H. Jensen. A graph-based genetic algorithm and generative model/monte carlo tree search for the exploration of chemical space. *Chemical Science*, 10(12):3567–3572, 2019. doi: 10.1039/c8sc05372c. URL https://doi.org/10.1039/c8sc05372c.

D. P. Kingma and J. Ba. Adam: A method for stochastic optimization. *CoRR*, abs/1412.6980, 2015.

D. P. Kingma and M. Welling. Auto-encoding variational bayes. *arXiv:1312.6114*, 2013.

T. Kluyver, B. Ragan-Kelley, F. Pérez, B. Granger, M. Bussonnier, J. Frederic, K. Kelley, J. Hamrick, J. Grout, S. Corlay, P. Ivanov, D. Avila, S. Abdalla, and C. Willing. Jupyter Notebooks – a publishing format for reproducible computational workflows. In F. Loizides and B. Schmidt, editors, *Positioning and Power in Academic Publishing: Players, Agents and Agendas*, pages 87–90. IOS Press, 2016.

M. Krenn, F. Häse, A. Nigam, P. Friederich, and A. Aspuru-Guzik. Self-referencing embedded strings (SELFIES): A 100% robust molecular string representation. *Machine Learning: Science and Technology*, 1(4):045024, Nov. 2020. doi: 10.1088/2632-2153/aba947. URL https://doi.org/10.1088/2632-2153/aba947.

M. J. Kusner, B. Paige, and J. M. Hernández-Lobato. Grammar variational autoencoder. *1703.01925*, 2017.

G. Landrum et al. RDKit: Open-source cheminformatics. http://www.rdkit.org.

Q. Liu, M. Allamanis, M. Brockschmidt, and A. L. Gaunt. Constrained graph variational autoencoders for molecule design. *The Thirty-second Conference on Neural Information Processing Systems*, 2018.

W. McKinney. Data Structures for Statistical Computing in Python. In *Python in Science Conference*, pages 56–61, Austin, Texas, 2010. doi: 10.25080/Majora-92bf1922-00a.

A. Nigam, P. Friederich, M. Krenn, and A. Aspuru-Guzik. Augmenting genetic algorithms with deep neural networks for exploring the chemical space. In *International Conference on Learning Representations*, 2020. URL `https://openreview.net/forum?id=H1lmyRNFvr`.

A. Nigam, R. Pollice, M. Krenn, G. dos Passos Gomes, and A. Aspuru-Guzik. Beyond generative models: Super-fast traversal, optimization, novelty, exploration and discovery (STONED) algorithm for molecules using SELF-IES. *Chemrxiv preprint*, Jan. 2021. doi: 10.26434/chemrxiv.13383266.v2. URL `https://doi.org/10.26434/chemrxiv.13383266.v2`.

A. Paszke, S. Gross, F. Massa, A. Lerer, J. Bradbury, G. Chanan, T. Killeen, Z. Lin, N. Gimelshein, L. Antiga, A. Desmaison, A. Kopf, E. Yang, Z. DeVito, M. Raison, A. Tejani, S. Chilamkurthy, B. Steiner, L. Fang, J. Bai, and S. Chintala. Pytorch: An imperative style, high-performance deep learning library. In H. Wallach, H. Larochelle, A. Beygelzimer, F. d'Alché-Buc, E. Fox, and R. Garnett, editors, *Advances in Neural Information Processing Systems 32*, pages 8024–8035. Curran Associates, Inc., 2019. URL `http://papers.neurips.cc/paper/9015-pytorch-an-imperative-style-high-performance-deep-learning-library.pdf`.

F. Pedregosa, G. Varoquaux, A. Gramfort, V. Michel, B. Thirion, O. Grisel, M. Blondel, P. Prettenhofer, R. Weiss, V. Dubourg, J. Vanderplas, A. Passos, D. Cournapeau, M. Brucher, M. Perrot, and E. Duchesnay. Scikit-learn: Machine Learning in Python. *Journal of Machine Learning Research*, 12:2825–2830, 2011.

P. G. Polishchuk, T. I. Madzhidov, and A. Varnek. Estimation of the size of drug-like chemical space based on GDB-17 data. *Journal of Computer-Aided Molecular Design*, 27(8):675–679, Aug. 2013. doi: 10.1007/s10822-013-9672-4. URL `https://doi.org/10.1007/s10822-013-9672-4`.

D. Rogers and M. Hahn. Extended-connectivity fingerprints. *Journal of Chemical Information and Modeling*, 50(5): 742–754, Apr. 2010. doi: 10.1021/ci100050t. URL `https://doi.org/10.1021/ci100050t`.

A. Ronacher. Pallets/click. The Pallets Projects, Jan. 2021.

B. Sanchez-Lengeling and A. Aspuru-Guzik. Inverse molecular design using machine learning: Generative models for matter engineering. *Science*, 361(6400):360–365, July 2018. doi: 10.1126/science.aat2663. URL `https://doi.org/10.1126/science.aat2663`.

SciPy 1.0 Contributors, P. Virtanen, R. Gommers, T. E. Oliphant, M. Haberland, T. Reddy, D. Cournapeau, E. Burovski, P. Peterson, W. Weckesser, J. Bright, S. J. van der Walt, M. Brett, J. Wilson, K. J. Millman, N. Mayorov, A. R. J. Nelson, E. Jones, R. Kern, E. Larson, C. J. Carey, İ. Polat, Y. Feng, E. W. Moore, J. VanderPlas, D. Laxalde, J. Perktold, R. Cimrman, I. Henriksen, E. A. Quintero, C. R. Harris, A. M. Archibald, A. H. Ribeiro, F. Pedregosa, and P. van Mulbregt. SciPy 1.0: Fundamental algorithms for scientific computing in Python. *Nat Methods*, 17(3):261–272, Mar. 2020. ISSN 1548-7091, 1548-7105. doi: 10.1038/s41592-019-0686-2.

A. Supady, V. Blum, and C. Baldauf. First-principles molecular structure search with a genetic algorithm. *Journal of Chemical Information and Modeling*, 55(11):2338–2348, Nov. 2015. doi: 10.1021/acs.jcim.5b00243. URL `https://doi.org/10.1021/acs.jcim.5b00243`.

M. Vogt and J. Bajorath. ccbmlib – a python package for modeling tanimoto similarity value distributions. *F1000Research*, 9:100, Mar. 2020. doi: 10.12688/f1000research.22292.2. URL `https://doi.org/10.12688/f1000research.22292.2`.

X. Yang, J. Zhang, K. Yoshizoe, K. Terayama, and K. Tsuda. ChemTS: an efficient python library for de novo molecular generation. *Science and Technology of Advanced Materials*, 18(1):972–976, Nov. 2017. doi: 10.1080/14686996.2017.1401424. URL `https://doi.org/10.1080/14686996.2017.1401424`.

N. Yoshikawa, K. Terayama, M. Sumita, T. Homma, K. Oono, and K. Tsuda. Population-based de novo molecule generation, using grammatical evolution. *Chemistry Letters*, 47(11):1431–1434, Nov. 2018. doi: 10.1246/cl.180665. URL `https://doi.org/10.1246/cl.180665`.

J. You, B. Liu, R. Ying, V. Pande, and J. Leskovec. Graph convolutional policy network for goal-directed molecular graph generation. In *Proceedings of the 32nd International Conference on Neural Information Processing Systems*, NIPS'18, page 6412–6422, Red Hook, NY, USA, 2018. Curran Associates Inc.

