# OpenReview forum: "A reproducibility study of "Augmenting Genetic Algorithms with Deep Neural Networks for Exploring the Chemical Space""
_ML_Reproducibility_Challenge/2020 — Reject_

### Official Review · AnonReviewer2 · 2021-03-01
**GA with DNN:**

**Rating:** 7
**Confidence:** 4

**Review:**

The author(s) provide very details in their report regarding the proposed algorithm and data set (ZINC) details, the report is 17 pages which I believe is too long. The model description is well defined. They provide every detail of the code and algorithms that have been used by other papers and provide the citation, but I believe it could be abstracted. The evaluation criteria; logP score were used to evaluate the optimization.

**Familiar With The Original Paper:**

I have not read the original paper

**Reproducibility Summary:**

Report has summary

---

### Official Review · AnonReviewer1 · 2021-03-01
**properly results with some very minor issues.**

**Rating:** 10
**Confidence:** 5

**Review:**

In the very early sentences of your review paper, In line 4, "an adaptive, neural network-based, penalty that is supposed to" it's better to put a comma, right after the ***penalty***, "an adaptive, neural network-based penalty, that is supposed to." The link you prepared in line 69 as your GitHub repository, did not work properly.
In my opinion, it was better to test your algorithm with other optimizers rather than Adam, and other Loss functions except for binary cross-entropy. It helps you obtain more accurate results by searching in different conditions.
In line 94, you made a typo and wrote ***pratical*** instead of ***practical***.
What is the y-axis in Fig1 plots?! What does it refer to?
You'd better add Legend to each of your images, even if you're partializing them to a) b) c).

**Familiar With The Original Paper:**

I have not read the original paper

**Reproducibility Summary:**

Report has summary

---

### Official Review · AnonReviewer3 · 2021-03-01
**A reproducibility study of “Augmenting Genetic Algorithms with Deep Neural Networks for Exploring the Chemical Space**

**Rating:** 7
**Confidence:** 4

**Review:**

Overall, my evaluation of the paper titled “A reproducibility study of “Augmenting Genetic Algorithms with Deep Neural Networks for Exploring the Chemical Space” is as below:


•	The authors provide a comprehensive summary of the objectives of the paper, its results, challenges, and contributions. (Reproducibility Summary: 10/10)
•	The authors provide the areas of the original paper where they have tried to reproduce the results and adhere to them along with the paper. (Scope of reproducibility: 10/10)
•	The code from the original paper has been used and modified for the hyperparameter search performed on the paper. Overall, there are enough documentation, and the code sounds clean. (Code: 10/10)
•	Scope of the communication with the authors of the original paper is mentioned (Communication with original authors: 10/10)
•	Effect of different hyperparameters that change the results has been studied in detail which complements the results of the original paper. (Hyperparameter Search: 10/10)
•	Although the authors have performed comprehensive hyperparameters search and have quantified the effect on the result, the level of novelty in the paper is limited. Maybe the authors could perform some study on the neural network-based penalty and see if they can add additional improvement to the original paper. (Ablation Study                                           6/10)
•	There are detailed discussions with graphs on the effect of each hyperparameter on the outcome. The authors have been very meticulous. Have examined the method in detail and have suggested valuable improvement to the original paper (Discussion on results 10/10)
•	The detailed illustration of the results is very helpful, but the authors do not provide any recommendation on future work for reproducibility. They mainly summarized what they have tried and the differences in the outcome with respect to the original paper.  (Recommendations for reproducibility 5/10)
•	I did not recognize significant novel contributions from this paper. (Results beyond the paper                          5/10)
•	The paper is organized, coherent, clear, and well structured. (Overall organization and clarity 10/10)



**Familiar With The Original Paper:**

I have read the original paper

**Reproducibility Summary:**

Report has summary

---

### Decision · Program_Chairs · 2021-03-31

**Decision:**

Reject

**Comment:**

While the paper carried out a fair attempt at reproducing the results of the original paper, the lack of in-depth analysis and comparison of their results with those of the original paper are lacking.